# Plasma FGF21 Levels Are Not Associated with Weight Loss or Improvements in Metabolic Health Markers upon 12 Weeks of Energy Restriction: Secondary Analysis of an RCT

**DOI:** 10.3390/nu14235061

**Published:** 2022-11-28

**Authors:** Anouk Gijbels, Sophie Schutte, Diederik Esser, Charlotte C. J. R. Michielsen, Els Siebelink, Monica Mars, Marco Mensink, Lydia A. Afman

**Affiliations:** Division of Human Nutrition and Health, Department of Agrotechnology and Food Sciences, Wageningen University & Research, Stippeneng 4, 6708 WE Wageningen, The Netherlands

**Keywords:** FGF21, fibroblast growth factor 21, overweight, obesity, caloric restriction, energy restriction, weight loss, dietary intervention, liver fat, metabolic health

## Abstract

Recent studies suggest that circulating fibroblast growth factor 21 (FGF21) may be a marker of metabolic health status. We performed a secondary analysis of a 12-week randomized controlled trial to investigate the effects of two energy restriction (ER) diets on fasting and postprandial plasma FGF21 levels, as well as to explore correlations of plasma FGF21 with metabolic health markers, (macro)nutrient intake and sweet-taste preference. Abdominally obese subjects aged 40–70 years (*n* = 110) were randomized to one of two 25% ER diets (high-nutrient-quality diet or low-nutrient-quality diet) or a control group. Plasma FGF21 was measured in the fasting state and 120 min after a mixed meal. Both ER diets did not affect fasting or postprandial plasma FGF21 levels despite weight loss and accompanying health improvements. At baseline, the postprandial FGF21 response was inversely correlated to fasting plasma glucose (*ρ* = −0.24, *p* = 0.020) and insulin (*ρ* = −0.32, *p* = 0.001), HOMA-IR (*ρ* = −0.34, *p* = 0.001), visceral adipose tissue (*ρ* = −0.24, *p* = 0.046), and the liver enzyme aspartate aminotransferase (*ρ* = −0.23, *p* = 0.021). Diet-induced changes in these markers did not correlate to changes in plasma FGF21 levels upon intervention. Baseline higher habitual polysaccharide intake, but not mono- and disaccharide intake or sweet-taste preference, was related to lower fasting plasma FGF21 (*p* = 0.022). In conclusion, we found no clear evidence that fasting plasma FGF21 is a marker for metabolic health status. Circulating FGF21 dynamics in response to an acute nutritional challenge may reflect metabolic health status better than fasting levels.

## 1. Introduction

Fibroblast growth factor 21 (FGF21) is a peptide hormone that regulates metabolic homeostasis [1,2]. The exact biological functions and regulation of FGF21 vary widely between tissues and are not completely understood [1,2]. Circulating FGF21 levels are typically elevated in conditions of impaired metabolic health, such as obesity, non-alcoholic fatty liver disease (NAFLD) and type 2 diabetes (T2DM) [3,4]. Accordingly, circulating FGF21 levels have been reported to positively associate with BMI, body fat, liver fat content, blood pressure, insulin resistance (IR) and atherogenic lipid profiles [5,6], as well as predict incident metabolic syndrome and T2DM [7,8]. It is unclear whether and how weight loss and accompanying health improvements affect FGF21 levels, with some trials reporting reduced levels of circulating FGF21 upon diet-induced weight loss [9,10,11,12,13], and others reporting no effects on FGF21 [14,15,16,17,18,19,20].

Recently, FGF21 has been implicated in the metabolic adaptations to both under- and overfeeding. In the DIETFITS trial, fasting FGF21 level was associated with diet-induced weight loss, with higher FGF21 levels at baseline being associated with larger weight loss in response to a 12-month low-fat or low-carbohydrate diet [20,21]. Furthermore, in a 6-week trial, the change in circulating FGF21 upon energy restriction was associated with weight loss, with individuals with larger increases in fasting FGF21 achieving greater weight loss [22]. In addition, in a short-term trial, greater increases in fasting plasma FGF21 in response to a 1-day hypercaloric low-protein, high-fat diet were associated with larger weight loss in free-living conditions after 6 months [23]. Plasma FGF21 (response) thus seems to be related to the ability to lose weight.

Next to its metabolic effects, FGF21 also appears to play a role in the regulation of nutrient intake via central nervous system (CNS) signaling. Multiple large human genome-wide association studies (GWAS) have reported associations between *FGF21* gene variants and carbohydrate, sugar, and alcohol consumption and sweet-taste preference [24,25,26,27]. Furthermore, higher fasting FGF21 levels have been associated with lower sweet-taste preference [26] and lower soda consumption [28]. Animal studies suggest that in response to carbohydrate and in particular sugar intake, FGF21 is induced in the liver, enters the circulation and subsequently acts as negative feedback signal via the liver-brain-axis to suppress further sweet-taste preference and carbohydrate intake [29,30,31]. As of yet, evidence from human studies on the link between plasma FGF21 and nutrient intake and taste preference is sparse.

We previously reported the effects of two 12-week energy restriction (ER) diets differing in nutrient quality in overweight and obese adults [32]. Both ER diets resulted in substantial weight loss and concomitant improvements in various health parameters, including reduced liver fat, and decreased fasting glucose and insulin levels. In the present study, we investigated whether these diet-induced changes in body weight and related health outcomes were accompanied by changes in fasting and postprandial FGF21 levels. Additionally, we explored correlations of plasma FGF21 with markers of metabolic health, weight loss, habitual (macro)nutrient intake and sweet-taste preference.

## 2. Materials and Methods

The present study is a secondary analysis of a 12-week parallel randomized controlled trial (RCT) that was performed at Wageningen University, the Netherlands, in 2014. The aim of this trial was to investigate the effects of two different ER diets on weight loss and cardiometabolic health outcomes. Details have been described previously [32]. 

### 2.1. Participants and Study Design

The study population consisted of 110 women and men aged 40–70 years with abdominal obesity (BMI > 27 kg/m^2^ or waist circumference > 88 cm for women or >102 cm for men). Exclusion criteria were diagnosis of diabetes, heavy alcohol consumption, smoking, unstable body weight, diagnosis of long-term medical condition, and use of medication that is known to interfere with glucose or lipid metabolism. 

Eligible subjects were randomly assigned to one of three intervention groups: a 25% ER high-nutrient-quality diet (*n* = 40), a 25% ER low-nutrient-quality diet (*n* = 40), or a control group (*n* = 30). At baseline and after 12 weeks of intervention, participants visited hospital Gelderse Vallei (Ede, the Netherlands) for an MRI scan and—on a separate day—visited Wageningen University after an overnight fast for measurements of general health and blood metabolites in the fasting state, as well as in response to a mixed meal.

### 2.2. Dietary Intervention

The dietary intervention strategy has been described in detail previously [32]. Briefly, both the ER diets were energy restricted by 25% of the estimated energy requirement of each participant. The high-nutrient-quality diet (HQ) was designed to improve metabolic health and contained an increased amount of soy protein, fiber, monounsaturated fat (MUFA), and omega-3 fatty acids. The low-nutrient-quality diet (LQ) contained an increased amount of saturated fat (SFA), animal protein, and fructose. The control group did not receive an intervention and was instructed to maintain their usual dietary habits. Participants in the ER groups received dietary advice and key food products. Adherence to the ER diets was assessed based on participants’ food diaries, taking reported deviations from the dietary advice and leftover key food products into account [32].

### 2.3. General Health Measures

Systolic and diastolic blood pressure were measured automatically for 10 min with a 3-min interval using a DINAMAP PRO100. After an overnight fast, blood samples were collected from an intravenous cannula before and 30, 60, 120, 180, 240, and 360 min after consumption of a liquid mixed meal (3833 kJ; 76.3 g carbohydrates, 17.6 g protein, 60 g fat). To estimate IR, we used the homeostatic model assessment of insulin resistance (HOMA-IR), which is calculated as (fasting glucose [mmol/L] × fasting insulin [mU/L])/22.5 [33]. IR was defined as HOMA-IR > 2.5 [34]. Incremental area under the curves (iAUC) for glucose and insulin were calculated using the trapezoid method.

### 2.4. Plasma FGF21

Plasma FGF21 levels were measured in plasma samples from the fasting state and 120 min after consumption of the mixed meal using ELISA, according to manufacturer’s instructions (FGF-21 Human ELISA Kit, ab125966, Abcam, Cambridge, UK). The inter- and intra-assay coefficients of variation were 12.4% and 5.9%, respectively. All samples from two subjects and one sample from another subject fell outside of the range of the standard curve and were therefore excluded from analyses.

### 2.5. Intra-Hepatic Lipid Content and Abdominal Fat Distribution

Intra-hepatic lipid content (IHL) was quantified using proton magnetic spectroscopy (^1^H-MRS) on a 3T whole-body scanner (Siemens, Munich, Germany). Details have been described previously [32]. MRS spectra were analyzed using jMRUI software v5.2. NAFLD was defined as IHL > 5.56% [35]. Abdominal fat distribution was evaluated as subcutaneous (SAT) and visceral adipose tissue (VAT) areas in the abdomen and assessed using magnetic resonance imaging (MRI). SAT and VAT were quantified in a single-slice transverse image at the inter-vertebral space L3-L4 using the semi-automatic software program HippoFatTM [36].

### 2.6. Clinical Chemistry

Plasma glucose, insulin, and triglycerides, as well as serum total cholesterol, HDL cholesterol and safety parameters of liver function (aminotransferase [ALAT], aspartate aminotransferase [ASAT], and gamma-glutamyl transferase [γGT]) were analyzed photometrically (Cobas 8000, Roche Diagnostic Limited, Switzerland) by a center for medical diagnostics (Stichting Huisartsenlaboratorium Oost, Velp, the Netherlands). Plasma free fatty acids (FFA) were determined with an enzymatic assay (INstruchemie, Delfzijl, The Netherlands). HbA1c was determined in whole blood by hospital Gelderse Vallei (Ede, The Netherlands). 

### 2.7. Habitual Dietary Intake and Sweet-Taste Preference

Habitual dietary intake including alcohol consumption was assessed by a validated 131-item semi-quantitative food frequency questionnaire (FFQ) [37,38]. Preference for sweet foods was assessed by a digital food preference ranking task [39]. Sweet-taste preference scores range from 1.5 to 3.5, with higher scores indicating greater preference for sweet foods compared to savory foods.

### 2.8. Statistical Analyses

Normality of variables was visually inspected using residual Q-Q plots. Skewed variables (plasma FGF21, VAT/SAT ratio, IHL, iAUC plasma glucose, fasting plasma insulin, iAUC plasma insulin, fasting plasma triglycerides, fasting plasma FFA, HOMA-IR, ASAT, and γGT) were log transformed (log2) to improve normality. Treatment effects on fasting FGF21 levels were analyzed using a general linear model for univariate analysis (ANCOVA), with baseline FGF21 level as covariate. Treatment effects on the postprandial FGF21 response were analyzed using ANOVA with the change in postprandial response as dependent variable. Correlations between variables were tested using Spearman’s correlation coefficient. Differences in plasma FGF21 levels between tertiles of nutrient intake were analyzed using ANCOVA with adjustment for gender, and LSD post-hoc testing was used if overall differences were statistically significant. To evaluate the robustness of the results, we performed sensitivity analyses excluding outliers. Outliers were defined as plasma FGF21 levels that deviated more than two standard deviations (SD) from the mean. Data analysis was performed using IBM SPSS Statistics version 25.0 (IBM Corp., Armonk, NY, US). Two-tailed *p* < 0.05 was considered statistically significant.

## 3. Results

A total of 100 subjects completed the study (*n* = 6 drop-outs in HQ, *n* = 1 drop-out in LQ, and *n* = 3 drop-outs in the control group [32]). Plasma FGF21 data were available from 98 subjects. Approximately half of the study population were women, median age was 62 years and median BMI was 30.8 kg/m^2^. Baseline demographics and clinical characteristics were similar across intervention groups (Table 1).

In five participants (*n* = 2 in HQ, *n* = 1 in LQ, and *n* = 2 in the control group), plasma FGF21 levels deviated more than 2 SD from the mean both in the fasting and postprandial state, and both at baseline and after 12 weeks of intervention. Another two participants (*n* = 1 in HQ and *n* = 1 in LQ) had deviating plasma FGF21 levels in the postprandial state at baseline or in the fasting state after 12 weeks of intervention, respectively. 

The reported dietary intakes of participants on the ER diets were in agreement with the advised (macro)nutrient composition, with higher intakes of MUFA, polyunsaturated fat, plant-based protein, and fiber, and lower intakes of SFA and fructose in the high-nutrient-quality diet compared to the low-nutrient-quality diet [32].

### 3.1. Effects of Dietary Interventions on Plasma FGF21

As has been previously reported, 12 weeks of 25% ER resulted in substantial weight loss (mean ± SD: −8.4 ± 3.2 kg in HQ and −6.3 ± 3.9 kg in LQ) and concomitant improvements in markers of metabolic health, including more favorable abdominal fat distribution, and reductions in IHL, fasting glucose, and fasting insulin levels [32]. On average, plasma FGF21 levels were not affected by the interventions, nor did the interventions affect the postprandial FGF21 response (Table 2; Figure 1). Pooling of the ER groups together yielded similar results (ER: ∆ fasting FGF21 −0.11 ng/ml, 95% CI −0.20 to −0.01, *p* ER vs. control group = 0.29; ∆ postprandial FGF21 response −0.02 ng/ml, 95% CI −0.10 to 0.06, *p* ER vs. control group = 0.62). Exclusion of outliers did not affect the results (all *p* > 0.19).

### 3.2. Postprandial Plasma FGF21 Response at Baseline

On average, plasma FGF21 tended to decrease 120 min after consumption of the mixed meal (∆ −0.06 ng/ml, 95% CI −0.11 to 0.001, *p* = 0.055), although inter-individual variation in response was high (Figure 2A). Exclusion of outliers strengthened this effect (∆ −0.12 ng/ml, 95% CI −0.23 to −0.02, *p* = 0.026). 

### 3.3. Plasma FGF21 and Correlations with Markers of Metabolic Health

At baseline, fasting plasma FGF21 concentrations were inversely correlated to plasma FFA (*ρ* = −0.22, *p* = 0.03), and not correlated to BMI, IHL, HOMA-IR or any of the other assessed cardiometabolic parameters (Figure 2C and Appendix A; Appendix A). The change in plasma FGF21 from the fasting state to 120 min postprandially was inversely correlated to fasting plasma glucose, fasting insulin, HOMA-IR, VAT and ASAT (*ρ* = −0.34 to −0.23, all *p* < 0.05), but did not correlate with IHL or the postprandial response of plasma glucose or insulin (Figure 2C and Appendix A; Appendix A). Exclusion of outliers attenuated the inverse correlations of the postprandial FGF21 response with plasma glucose (*ρ* = −0.19, *p* = 0.069) and ASAT (*ρ* = −0.20, *p* = 0.051), annulled the correlation with VAT (*ρ* = −0.18, *p* = 0.14), and did not affect the other correlations.

To further explore the relationship between the postprandial FGF21 response and HOMA-IR, we stratified the total study population into insulin-sensitive (*n* = 44) and insulin-resistant (*n* = 53) subjects based on HOMA-IR ≤ 2.5 and >2.5 [38]. In insulin sensitive subjects, FGF21 did not change from fasting to 120 min postprandially, while in insulin resistant subjects, FGF21 declined 120 min after consumption of the mixed meal, also after adjustment for fasting FGF21, age, gender, and BMI (∆ −0.03 ng/ml, 95% CI −0.06 to 0.11 vs. ∆ −0.13 ng/ml, 95% CI −0.20 to −0.05; *p* = 0.012) (Figure 2B). Exclusion of outliers (*p* = 0.007) or additional adjustment for NAFLD status (IHL ≤ 5.56% or > 5.56% [39]) or IHL did not affect this result (*p* = 0.013 and *p* = 0.018, respectively).

Next to testing correlations at baseline, we also explored correlations between the change in fasting and postprandial FGF21 response and change in markers of metabolic health. In the high-nutrient-quality diet group, change in both fasting FGF21 and the postprandial FGF21 response upon the 12 weeks of intervention was inversely correlated to the reduction in abdominal SAT (*ρ* = −0.49, *p* = 0.012; *ρ* = −0.45, *p* = 0.023, respectively) (Figure 2C; Appendix A; Appendix A). Furthermore, in the high-nutrient-quality diet group, the change in postprandial FGF21 response was inversely correlated to the reduction in fasting plasma triglycerides (*ρ* = −0.36, *p* = 0.041) (Figure 2C; Appendix A). In the low-nutrient-quality diet, the change in postprandial FGF21 response was positively correlated to the reduction in fasting insulin levels (*ρ* = 0.35, *p* = 0.034) (Figure 2C; Appendix A). In the control group, changes in both fasting and postprandial FGF21 response were inversely correlated to change in HbA1c, and change in fasting FGF21 levels was positively correlated to change in fasting glucose levels (Figure 2C; Appendix A). These latter three correlations, however, were driven by data points from two participants, and exclusion of these data points resulted in a loss of significant correlations (*ρ* = −0.30 to 0.25, *p* > 0.13).

Exclusion of outliers resulted in positive correlations between the change in fasting FGF21 and the change in fasting insulin, iAUC insulin, and HOMA-IR in the high-nutrient-quality diet group (*ρ* = 0.36 to 0.51, all *p* < 0.05). In addition, exclusion of outliers attenuated the inverse correlation between the change in postprandial FGF21 response and change in fasting plasma triglycerides in the high-nutrient-quality diet group (*ρ* = −0.31, *p* = 0.097), annulled the positive correlation between change in postprandial FGF21 response and change in fasting insulin in the low-nutrient-quality diet group (*ρ* = 0.27, *p* = 0.11), and did not affect the other correlations.

### 3.4. Plasma FGF21 and Habitual (Macro)Nutrient Intake and Sweet-Taste Preference

Since FGF21 has been linked to (macro)nutrient intake and preference, we compared fasting plasma FGF21 levels between tertiles of habitual nutrient intake, alcohol consumption, and sweet-taste preference. Fasting FGF21 levels were lower in individuals in the highest tertile of habitual polysaccharide intake compared to individuals in lower tertiles of polysaccharide intake (overall *p* = 0.022; tertile 3 vs. tertile 1, *p* = 0.035; tertile 3 vs. tertile 2, *p* = 0.009; Figure 3C). FGF21 levels did not differ between individuals in tertiles of habitual intake of carbohydrates, protein, fat, mono- and disaccharides (sugars), or alcohol consumption (Figure 3). In addition, plasma FGF21 levels did not differ according to sweet-taste preference (Figure 3G). The ranges and means of the tertiles of habitual nutrient intake, alcohol consumption, and sweet-taste preference are reported in Appendix A. Exclusion of outliers attenuated the differences in fasting FGF21, according to habitual polysaccharide intake (overall *p* = 0.098; tertile 3 vs. tertile 1, *p* = 0.066; tertile 3 vs. tertile 2, *p* = 0.063).

### 3.5. Baseline Plasma FGF21 and Weight Loss

In an explorative analysis, we examined whether fasting plasma FGF21 and the FGF21 postprandial response at baseline were associated with weight loss after 12 weeks of ER intervention. Fasting plasma FGF21 levels at baseline were not correlated to weight loss in either of the ER groups (HQ: *ρ* = −0.01, *p* = 0.96; LQ: *ρ* = −0.27, *p* = 0.10; HQ and LQ combined: *ρ* = −0.08, *p* = 0.50 Figure 4A). The postprandial change in FGF21 at baseline was borderline positively associated with weight change in the low-nutrient-quality diet group (*ρ* = 0.30, *p* = 0.065) but not in the high-nutrient-quality diet group or in the two groups combined (HQ: *ρ* = −0.23, *p* = 0.19; HQ and LQ combined: *ρ* = 0.06, *p* = 0.63 Figure 4B). Exclusion of outliers resulted in an inverse correlation between fasting FGF21 at baseline and weight loss (*ρ* = −0.33, *p* = 0.048) and attenuated the correlation with the postprandial FGF21 response in the low-nutrient-quality diet group (*ρ* = 0.27, *p* = 0.10).

## 4. Discussion

In this study, we investigated the effects of two 12-week energy restriction (ER) diets differing in nutrient quality on fasting and postprandial plasma FGF21 levels, and explored correlations between plasma FGF21 and markers of metabolic health. Neither overall ER nor high vs. low nutrient quality of the ER diets affected circulating fasting or postprandial FGF21 levels. Diet-induced weight loss and liver fat reduction were not accompanied by changes in fasting or postprandial plasma FGF21. Fasting plasma FGF21 was not correlated to markers of metabolic health at baseline, but the postprandial FGF21 response to the mixed meal was inversely correlated to fasting glucose, insulin, HOMA-IR, visceral AT and the liver enzyme ASAT. In addition, we assessed associations between habitual dietary intake and circulating FGF21 levels. Fasting FGF21 levels were lowest in individuals with the highest intake of polysaccharides, but did not differ according to intake of mono- and disaccharides, alcohol consumption, or sweet-taste preference. 

Despite substantial weight loss of 7.3 kg on average and concomitant health improvements [32], 12 weeks of 25% energy reduction did not affect fasting FGF21 levels. Previous weight loss trials have reported conflicting findings, with some trials reporting a reduction in fasting FGF21 levels upon diet-induced weight loss [9,10,11,12,13], and others reporting an increase [40] or no effect on circulating FGF21 levels [14,15,16,17,18,19,20]. Generally, the studies that report a change in plasma fasting FGF21 were performed in individuals with more severely impaired metabolic health (e.g., T2DM, NAFLD, morbid obesity), who typically have elevated FGF21 levels and thus more room for improvement compared to individuals in this study. In addition, the degree of weight loss, as well as the content and composition of the intervention diets may contribute to disagreement between studies. Weight loss as a result of bariatric surgery, which is commonly larger than weight loss achieved by ER, such as in our study, is also not consistently accompanied by changes in FGF21 levels [41]. It thus seems that weight loss and accompanying health improvements do not consistently affect fasting FGF21 levels.

Although fasting FGF21 levels were not correlated to liver fat or markers of glucose or lipid metabolism at baseline, change in fasting plasma FGF21 levels upon the 12-week intervention was inversely correlated to change in SAT, with larger reductions in SAT being accompanied by an increase in fasting plasma FGF21. In contrast, after the exclusion of five outliers, the reductions in HOMA-IR and both fasting and postprandial plasma insulin upon the intervention were correlated to a decrease in fasting FGF21, indicating that change in fasting FGF21 may be a marker for improvement in insulin sensitivity. These correlations, however, were present only in the high-nutrient-quality diet group, and it is unclear why we did not find similar correlations in the low-nutrient-quality diet group, given that the reductions in SAT, plasma insulin, and HOMA-IR were comparable upon the two ER diets. In addition, excluding participants with deviating values from analysis may lead to bias and therefore, these results should be interpreted with caution. 

At baseline, the postprandial FGF21 response was inversely correlated to insulin resistance as estimated by HOMA-IR, with more insulin resistant individuals exhibiting a larger postprandial decline in plasma FGF21 compared to insulin sensitive individuals. Circulating FGF21 levels have previously been reported to modestly decrease for 1 to 4 h after ingestion of meals containing fat, protein, or a combination of macronutrients [13,42,43,44]. In line with our findings, individuals with T2DM have been reported to have a more pronounced decline in postprandial FGF21 levels after a mixed meal compared to individuals with normal glucose metabolism [13]. These observations suggest that circulating FGF21 concentrations in response to mixed meals may be less well-controlled in impaired metabolic health.

The regulation of fasting and postprandial FGF21 levels in response to different nutrients and nutrient combinations is complex and not fully understood [1,2]. Overall, plasma FGF21 levels appear to display a circadian rhythm, with peak levels during fasting and lower levels during feeding [45]. During fasting, liver-derived FGF21 is primarily regulated by the transcription factor PPARα, and intake of fat, protein or a mixed meal results in diminished PPARα-mediated FGF21 secretion, possibly (partly) via a reduction in plasma FFA levels [15,45,46]. High intake of pure simple sugars or alcohol forms an exception; these nutrients acutely elevate plasma FGF21 [30,47,48,49], likely via activation of hepatic ChREBP [29,50]. Similar to our findings of a more pronounced postprandial response to a mixed meal in IR, plasma FGF21 excursions after pure fructose or glucose ingestion have been found to be larger in individuals with metabolic syndrome compared to healthy individuals [47]. In addition, the postprandial FGF21 response to fructose has been positively correlated to measures of hepatic and adipose tissue IR [48]. Circulating FGF21 in response to acute nutritional challenges may thus be a marker of metabolic health. Various factors including insulin, glucagon, adiponectin, and FFA appear to be involved in the regulation of FGF21 [1]. Greater FGF21 excursions in response to acute (nutritional) challenges, i.e., poor control of plasma FGF21 levels, in conditions of impaired metabolic health may reflect the disturbed control of the signals that regulate FGF21. Further research into the mechanisms that underlie altered FGF21 dynamics in impaired metabolic health is warranted.

If the postprandial FGF21 response is a marker for metabolic health, it could be expected that the substantial weight loss and concomitant health improvements after 12 weeks of ER in this study would be accompanied by a reduction in the postprandial FGF21 response. We, however, found no effects of the 12-week ER interventions on postprandial FGF21. A trial in individuals with T2DM and morbid obesity did report that weight loss of ~6.5 kg upon a 3-week very-low-calorie diet was accompanied by a less pronounced decline in postprandial in FGF21 levels up to 3 h after a mixed meal [13]. Our null findings may be due to our relatively less metabolically impaired study population and therefore less pronounced health improvements compared to the study in T2DM patients. In addition, we observed great inter-individual variation in the postprandial FGF21 response, with on average a modest decrease in plasma FGF21 levels 120 min after the mixed meal. Still, in approximately one third of the subjects, plasma FGF21 did not change or increased postprandially, which highlights the complexity of plasma FGF21 regulation and the challenge of interpreting plasma FGF21 (dynamics).

Recently, FGF21 levels have been suggested to be predictive of weight loss. We, however, did not find baseline fasting nor postprandial FGF21 concentrations to correlate with weight loss upon 12 weeks of ER. After exclusion of five outliers, we did observe that higher baseline fasting FGF21 was correlated to larger weight loss in the low- and not the high-nutrient-quality diet group, although these results should be interpreted with caution. In the DIETFITS trial, a trial in which 609 overweight or obese adults were randomized to follow a low-fat or low-carbohydrate weight-loss diet for 3 months, higher baseline FGF21 levels were associated with larger weight loss [21]. The inconsistent findings in our study might be due to limited power. In addition, in the DIETFITS trial, participants were instructed to drastically reduce their carbohydrate or fat intake, while there were no specific instructions regarding caloric intake. As FGF21 has been suggested to regulate energy homeostasis in humans by amongst others decreasing caloric intake [1], the association between baseline FGF21 and weight loss may thus partly result from differences in FGF21-mediated caloric intake. In our study, participants were explicitly instructed and monitored to reduce caloric intake by 25%, which leaves less room for regulation of caloric intake by FGF21.

FGF21 has also been implicated in the regulation of macronutrient intake. We found that individuals with higher habitual intake of polysaccharides had lower fasting FGF21 levels, but we found no associations between FGF21 and sweet-taste preference or habitual intake of mono- and disaccharides. Previously, higher soda consumption has been associated with lower circulating FGF21 [28], and *FGF21* gene variants have been related to higher total carbohydrate intake [24,25,26,27]. The only GWAS study that made a distinction between type of carbohydrates [26], however, found that the association between *FGF21* gene variants and carbohydrate intake could be attributed solely to higher intake of mono- and disaccharides and not to higher intake of polysaccharides. Animal studies also indicate that FGF21 regulates intake of mono- and disaccharides specifically [29], so our finding of lower plasma FGF21 levels in individuals with high intake of polysaccharides remains unexplained. 

Strengths of this study include its RCT design, with—next to two ER dietary intervention groups—a control group that did not receive dietary advice, which reduces bias and confounding factors. In addition, this study had relatively few drop-outs and compliance to ER was high, as is demonstrated by the substantial weight loss that participants achieved. 

This was a secondary analysis of an RCT designed to study effects of ER diets differing in nutrient quality on metabolic health outcomes, and the original sample size calculation was based on the power to detect differences in IHL [32]. Given the large inter-individual variation in plasma FGF21 levels, this study may have been underpowered to detect effects on plasma FGF21. Further large trials are thus needed to confirm our findings. Another limitation may be that we measured total plasma FGF21 protein, rather than bioactive FGF21. In the circulation, FGF21 is rapidly degraded by enzymatic cleavage, rendering it inactive [51]. Measuring bioactive rather than total FGF21 concentrations may be more physiologically relevant. Furthermore, we only measured postprandial FGF21 levels at a single timepoint (2 h after consumption of the mixed meal), and do not know the plasma FGF21 levels in the time between. We, however, expect that plasma FGF21 steadily decreased throughout the 2 h after consumption, since a previous study with more frequent sampling times showed a continuous decline in plasma FGF21 up to 3 h after a mixed meal [13]. Lastly, we did not assess physical activity level and therefore cannot rule out that differences in physical activity level at baseline or follow-up may have affected plasma FGF21 levels, given that exercise modifies circulating FGF21 levels [52]. However, we consider it unlikely that the ER groups differed in physical activity level at baseline or follow-up, because (1) interventions were randomly allocated, (2) participants were unaware of the high-nutrient-quality vs. low-nutrient-quality distinction, and (3) participants were instructed to maintain their habitual physical activity level throughout the study.

## 5. Conclusions

In conclusion, weight loss and concomitant health improvements upon a 12-week 25% ER diet were not accompanied by changes in fasting or postprandial plasma FGF21 levels in middle-aged individuals with abdominal obesity. Neither overall ER nor nutrient quality affected plasma FGF21. In addition, we found no robust evidence that fasting plasma FGF21 is a marker for metabolic health status. We did find indications that circulating FGF21 dynamics in response to a nutritional challenge may reflect metabolic health status. FGF21′s metabolic regulation and functions are greatly complex and further research on the potential of circulating FGF21 dynamics as a marker of metabolic health is warranted.

## Figures and Tables

**Figure 1 nutrients-14-05061-f001:**
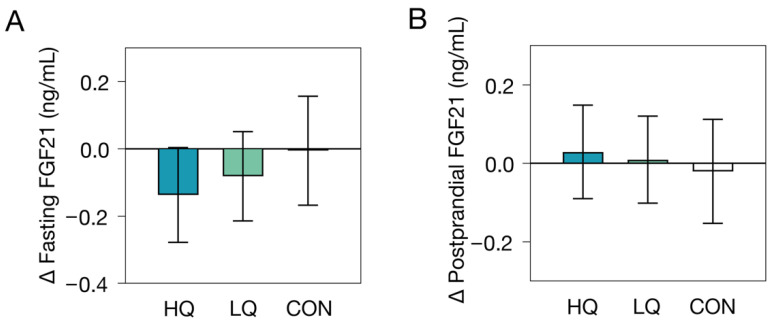
Effects of two 12-week energy restriction diets (HQ, high-nutrient-quality diet; LQ, low-nutrient-quality diet) or control group (CON) on fasting and postprandial plasma FGF21 levels. (**A**) Geometric mean (95% CI) change in fasting plasma FGF21 upon a 12-week diet, as tested by ANCOVA with adjustment for baseline fasting FGF21 levels. (**B**) Geometric mean (95% CI) change in postprandial FGF21 response (∆120–0 min) to a mixed meal upon a 12-week diet, as tested by univariate ANOVA.

**Figure 2 nutrients-14-05061-f002:**
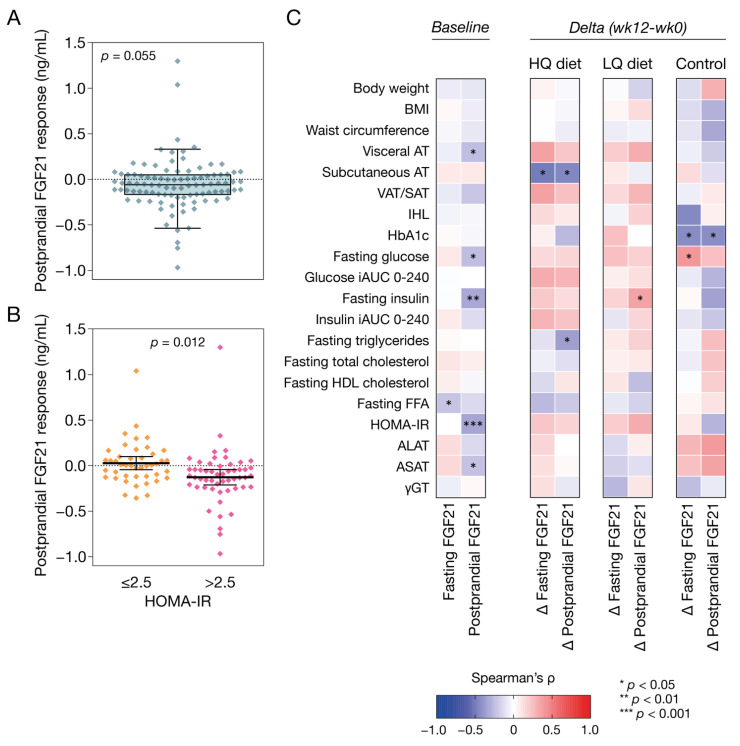
Postprandial FGF21 response at baseline and correlations between plasma FGF21 and markers of metabolic health. (**A**) Box plot with individual data points of the change in plasma FGF21 levels from the fasting state to 120 min after the consumption of the mixed meal in the complete population at baseline (*p* = 0.055, as tested using a paired-samples T-test). The box plot represents the 5th percentile, first quartile, median, third quartile, and 95th percentile. (**B**) Individual postprandial FGF21 responses according to baseline HOMA-IR: ≤2.5 or >2.5 (*p* = 0.012 for the difference between groups, as tested with ANCOVA, adjusted for fasting FGF21, age, gender, and BMI. (**C**) Spearman correlations between fasting FGF21 and the postprandial FGF21 response with cardiometabolic parameters at baseline (left) and in response to a 12-week intervention (right). Abbreviations: BMI, body mass index; AT, adipose tissue; IHL, intra-hepatic lipid content; HbA1c, glycated hemoglobin A1c; iAUC, incremental area under the curve; HDL, high-density lipoprotein; HOMA-IR, homeostatic model assessment of insulin resistance; ALAT, aminotransferase; ASAT, aspartate aminotransferase; γGT, gamma-glutamyl transferase.

**Figure 3 nutrients-14-05061-f003:**
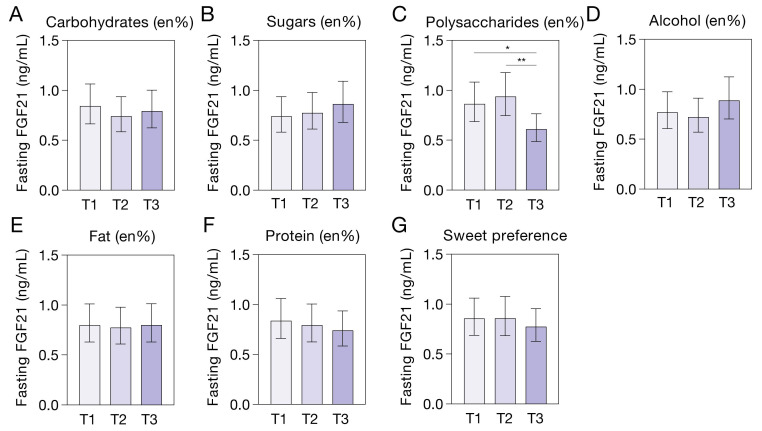
Fasting plasma FGF21 levels (adjusted geometric means with 95% CI) according to tertiles (T1: lowest tertile, T3: highest tertile) of habitual (macro)nutrient intake as % of daily energy intake (en%) (**A**–**C**,**E**,**F**), habitual alcohol consumption (**D**), and sweet-taste preference (**G**). FGF21 levels were lower in the highest tertile of polysaccharide intake compared to the lower tertiles (overall *p* = 0.022; tested by ANCOVA with adjustment for gender and LSD post-hoc testing; * *p* = 0.035; ** *p* = 0.009) and did not differ between tertiles of other nutrient intakes.

**Figure 4 nutrients-14-05061-f004:**
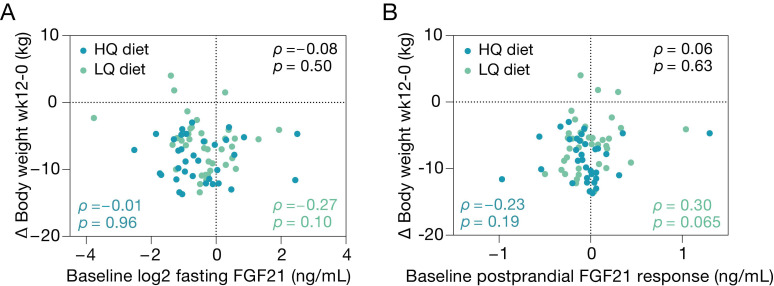
Scatter plots of fasting plasma FGF21 levels (**A**) and the postprandial FGF21 response (**B**) at baseline with weight change after 12 weeks of intervention on the high-nutrient-quality diet (HQ, dark green) and low-nutrient-quality diet (LQ, light green). Correlations were tested using Spearman correlation coefficients (LQ and HQ group combined in black).

**Table 1 nutrients-14-05061-t001:** Baseline characteristics of study participants with available FGF21 data.

	High-Nutrient- Quality Diet (*n* = 34)	Low-Nutrient- Quality Diet (*n* = 38)	Control Group (*n* = 26)
Women, *n* (%)	18 (52.9%)	21 (55.3%)	13 (50.0%)
Age, years	62 (53,65)	64 (54, 65)	61 (56, 66)
BMI, kg/m^2^	31.1 (28.7, 33.8)	30.8 (28.9, 33.4)	30.3 (28.1, 32.6)
Intra-hepatic lipid content, %	3.1 (1.4, 10.2)	4.9 (2.6, 9.7)	3.7 (2.1, 8.4)
NAFLD, *n* (%)	9 (26.5%)	14 (36.8%)	7 (26.9%)
HbA1c, mmol/mol	37 ± 3	36 ± 2	35 ± 3
HOMA-IR	2.7 (2.0, 4.9)	2.8 (1.6, 5.0)	2.6 (1.9, 4.4)
HOMA-IR > 2.5, *n* (%)	19 (55.9%)	21 (55.3%)	14 (53.8%)
Plasma glucose, mmol/L	5.7 ± 0.5	5.6 ± 0.7	5.7 ± 0.4
Plasma insulin, mU/L	10.9 (8.2, 17.5)	11.6 (7.0, 18.6)	10.9 (7.9, 17.0)
Plasma triglycerides, mmol/L	1.5 (1.1, 1.9)	1.7 (1.3, 2.3)	1.7 (1.4, 2.1)
Serum total cholesterol, mmol/L	5.5 ± 0.8	5.8 ± 1.0	5.5 ± 1.0
Serum HDL cholesterol, mmol/L	1.4 ± 0.4	1.3 ± 0.4	1.3 ± 0.4
Plasma free fatty acids, mmol/L	0.5 (0.4, 0.6)	0.4 (0.3, 0.5)	0.4 (0.3, 0.5)
Systolic blood pressure, mmHg	131 ± 15	126 ± 19	126 ± 14
Diastolic blood pressure, mmHg	76 ± 9	72 ± 8	75 ± 9
Alanine aminotransferase, U/L	24 (19, 33)	24 (20, 33)	25 (20, 33)
Aspartate aminotransferase, U/L	22 (19, 26)	22 (19, 25)	24 (20, 32)
Gamma-glutamyl transferase, U/L	23 (18, 33)	26 (18, 36)	23 (17, 32)

Data are presented as mean ± SD or median (25th percentile, 75th percentile). Abbreviations: BMI, body mass index; NAFLD, non-alcoholic fatty liver disease; HbA1c, glycated hemoglobin A1c; HOMA-IR, homeostatic model assessment of insulin resistance; HDL, high-density lipoprotein.

**Table 2 nutrients-14-05061-t002:** Change in fasting plasma FGF21 and the postprandial FGF21 response upon a 12-week energy restriction diet (low or high nutrient quality) compared to a control group.

	Baseline ^a^	Change after 12 Wks ^b^	*p*-Value ^c^
Fasting FGF21 (ng/mL)			
Control group (*n* = 26)	0.95 ± 1.86	−0.01 (−0.17, 0.16)	0.48
Low-Nutrient-Quality Diet (*n* = 38)	0.77 ± 1.89	−0.08 (−0.21, 0.05)	
High-Nutrient-Quality Diet (*n* = 34)	0.71 ± 1.92	−0.14 (−0.28, 0.004)	
Postprandial FGF21 (ng/mL)			
Control group (*n* = 26)	0.82 ± 2.00	0.00 (−0.15, 0.15)	0.57
Low-Nutrient-Quality Diet (*n* = 37)	0.71 ± 1.93	−0.09 (−0.21, 0.04)	
High-Nutrient-Quality Diet (*n* = 32)	0.64 ± 1.88	−0.10 (−0.24, 0.04)	
Postprandial FGF21 response (ng/mL)			
Control group (*n* = 26)	−0.12 ± 0.22	−0.02 (−0.15, 0.11)	0.86
Low-Nutrient-Quality Diet (*n* = 37)	−0.02 ± 0.27	0.01 (−0.10, 0.12)	
High-Nutrient-Quality Diet (*n* = 32)	−0.05 ± 0.34	0.03 (−0.09, 0.15)	

^a^ Values are geometric means ± SD (fasting and postprandial FGF21) or means ± SD (postprandial FGF21 response). ^b^ Values are (adjusted) means with 95% confidence intervals. ^c^ Differences between the groups were analyzed using ANCOVA with baseline FGF21 levels as covariate (fasting and postprandial FGF21) or using univariate ANOVA (postprandial FGF21 response).

## Data Availability

Data presented in this manuscript are available upon reasonable request.

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
