# Peer review of "Plasma FGF21 Levels Are Not Associated with Weight Loss or Improvements in Metabolic Health Markers upon 12 Weeks of Energy Restriction: Secondary Analysis of an RCT"

_nutrients, 2022, doi:10.3390/nu14235061_

Round 1
Reviewer 1 Report
Thanks for conducting this important research to help establish the utility of FGF21 hormone as a regulator of metabolic homeostasis. Please find below these minor suggestions for you to address.
1. Line 15 and 150: You enrolled 110 subjects, but the results were based on 100 subjects. Please clearly describe this and add the attrition rate in the methods section. You mentioned low attrition as a strength of the study. So, it will help if you include the exact attrition rate you were referring to in your methods.
2. Line 138 and 139. Please add the list of the variables that needed log transformation here.
3. Your study duration was 12 wks. Don’t you think this can be a study limitation given the fact that your study participants were mild to moderate insulin resistant? If you agree with my suggestion…then add this study limitation. If you do not agree, then provide a discussion point to justify why you think 12 weeks was adequate for you to have detected changes in fasting and postprandial FGF21 levels.
Author Response
We would like to thank the reviewer for the valuable comments and the positive feedback on our manuscript. We have addressed the comments point-by-point below.
- Line 15 and 150: You enrolled 110 subjects, but the results were based on 100 subjects. Please clearly describe this and add the attrition rate in the methods section. You mentioned low attrition as a strength of the study. So, it will help if you include the exact attrition rate you were referring to in your methods.
We have added the attrition rates per intervention group in lines 150-152:
“A total of 100 subjects completed the study (n = 6 drop-outs in the high-nutrient-quality diet group, n = 1 drop-out in the low-nutrient-quality-diet group, and n = 3 drop-outs in the control group [REF]).”
- Line 138 and 139. Please add the list of the variables that needed log transformation here.
We have added the list of variables that we log transformed in lines 137-139:
“Skewed variables (plasma FGF21, VAT/SAT ratio, IHLC, iAUC plasma glucose, fasting plasma insulin, iAUC plasma insulin, fasting plasma triglycerides, fasting plasma FFA, HOMA-IR, ASAT, and γGT) were log transformed (log2) to improve normality.”
- Your study duration was 12 wks. Don’t you think this can be a study limitation given the fact that your study participants were mild to moderate insulin resistant? If you agree with my suggestion…then add this study limitation. If you do not agree, then provide a discussion point to justify why you think 12 weeks was adequate for you to have detected changes in fasting and postprandial FGF21 levels.
Indeed, our study participants were mild to moderate insulin resistant and overall had mild to moderately impaired metabolic health. About 40% could be considered to have NAFLD according to liver fat >5.5%. We think that the magnitude of weight loss and/or health improvements plays a bigger role in affecting plasma FGF21 than the study duration itself, since lifestyle changes can affect plasma FGF21 already within days or weeks (doi: 10.1371/journal.pone.0038022; doi: 10.1016/j.molmet.2018.03.010.). The energy restriction was quite substantial and therefore also resulted in substantial weight loss of 7.3 kg on average, which was accompanied by a 29% relative reduction in visceral adipose tissue, 19% relative reduction in liver fat, and 16% relative reduction in HOMA-IR. It could be speculated that perhaps we would have found changes in plasma FGF21 if the study had ran for a longer period or if the energy restriction had been larger, and thus weight loss and health improvements had been larger. We were however interested whether this moderate yet substantial weight loss with accompanying health improvements would affect plasma FGF21 levels, and we conclude that weight loss per se does not seem to affect plasma FGF21. Therefore, we do not regard the 12-week duration of the study as a limitation. We discuss in lines 284-290 that the fact that our study population was relatively healthy compared to some other studies may explain the discrepancy in findings between studies.

Reviewer 2 Report
In this secondary analysis of a previously completed randomized control trial, Gijbels et al. examined the relationship between fasting and post-prandial FGF21 levels in adults with obesity before and after a 12-week energy-restricted diet. The authors reported that FGF21 response to energy-restricted dietary intervention, but not fasting FGF21 level, could be a marker of metabolic health. Overall, this is a well-written manuscript. I would like the authors to consider a few extra analyses to ensure that their conclusions hold up for different subsets of patients.
I believe the introduction can be revised/shortened. Some background information can be moved to the discussion section to avoid distraction from the main focus.
As mentioned in the manuscript, FGF21 level is associated with obesity, insulin resistance/type 2 diabetes, and NAFLD. The authors used continuous variables for weight/BMI, insulin/ glucose/HOMA-IR, and IHLC. An attempt was made to stratify patients based on HOMA-IR (i.e.,=<2.5 or >2.5) but not for other markers (likely due to lack of correlation). But, previous reports suggest distorted physiology of FGF21 levels in pathological states such as NAFLD. Please consider examining correlations (of FGF21 and other markers) in individuals with NAFLD or use NAFLD status as a covariate in your analyses. Consider examining the HOMA-IR vs. FGF21 relationship in subjects with or without NAFLD.
Why was HOMA-IR 2.5 selected as the cut-off? Is this an arbitrary choice, or does this threshold predict the future risk of type 2 diabetes?
Was any physical activity data collected in the original study? If so, were the groups compared concerning their PA level at baseline or at follow-up? Clarifying this might be important to ensure that the differences in FGF21 dynamics (or lack thereof) are not affected by physical activity. If no data is collected, please briefly discuss this in the weaknesses.
It is interesting to see a negative association between FGF21 levels (fasting and post-prandial) and markers of glucose metabolism in the control group. What does the author think about this finding?
In subsection 3.4. Plasma FGF21 and habitual... Please make sure the figure names in the text and figure match.
Optional: I think it would be nice to include some information regarding regulating FGF21 synthesis/secretion by nutrients at the cellular level. This may help readers better understand the relationship between FGF21 levels and nutrient intake/sweet-taste preference.
Author Response
We thank the reviewer for the valuable comments and the extensive feedback on our manuscript. We have addressed the comments point-by-point below.
I believe the introduction can be revised/shortened. Some background information can be moved to the discussion section to avoid distraction from the main focus.
We shortened the introduction by summarizing the part on animal studies in lines 57-60.
As mentioned in the manuscript, FGF21 level is associated with obesity, insulin resistance/type 2 diabetes, and NAFLD. The authors used continuous variables for weight/BMI, insulin/ glucose/HOMA-IR, and IHLC. An attempt was made to stratify patients based on HOMA-IR (i.e.,=<2.5 or >2.5) but not for other markers (likely due to lack of correlation). But, previous reports suggest distorted physiology of FGF21 levels in pathological states such as NAFLD. Please consider examining correlations (of FGF21 and other markers) in individuals with NAFLD or use NAFLD status as a covariate in your analyses. Consider examining the HOMA-IR vs. FGF21 relationship in subjects with or without NAFLD.
We indeed performed an exploratory analysis with stratification for HOMA-IR because we observed a correlation between HOMA-IR and the postprandial FGF21 response and not for other variables due to a lack of correlation. We agree that exploring effect modification by NAFLD status is of interest and therefore performed additional analyses:
- Fasting FGF21 (p=0.88) and the postprandial FGF21 response (p=0.63) did not differ between non-NAFLD (n=49) and NAFLD (n=30).
- Correlations between HOMA-IR and the postprandial FGF21 response were comparable between non-NAFLD and NAFLD individuals: ρ = -0.29, p = 0.043 and ρ = -0.45, p = 0.012, respectively.
- In addition, when stratifying for NAFLD status, the inverse correlations between the postprandial FGF21 response and fasting insulin were maintained, but correlations with fasting glucose, VAT, and ASAT were only present in the NAFLD, but not the non-NAFLD group.
- When stratifying for NAFLD status, the correlation between fasting FGF21 and plasma FFA is lost. In addition, VAT is positively correlated to fasting FGF21 in the non-NAFLD group and VAT/SAT ratio inversely correlated to fasting FGF21 in the NAFLD group.
- Additional adjustment for NAFLD status or IHLC did not affect the difference found in postprandial FGF21 between HOMA-IR ≤ 2.5 and > 2.5 (p = 0.013 and p = 0.018, respectively).
The loss of sign. correlations with postprandial FGF21 response in the non-NAFLD may be (partly) due to the smaller range in glucose/VAT/ASAT compared to the NAFLD group. As can be expected, the gender balance differs between the non-NAFLD and NAFLD groups (60% women vs. 40% women). This difference in gender balance may also affect the differences in correlations with FGF21 between the NAFLD and non-NAFLD groups. Ideally, we would additionally adjust or stratify for gender, but the relatively limited sample size does not allow that.
Importantly, the inverse correlations between postprandial FGF21 response and HOMA-IR (and insulin) are independent of NAFLD status. We added the outcomes of additional adjustment for NAFLD status/IHLC in the analyses of postprandial FGF21 response in HOMA-IR ≤ 2.5 vs. > 2.5 to the manuscript in lines 215-217:
“Additional adjustment for NAFLD status (IHLC ≤ 5.56% or > 5.56% [8]) or IHLC did not affect this outcome (p = 0.013 and p = 0.018, respectively).”
Why was HOMA-IR 2.5 selected as the cut-off? Is this an arbitrary choice, or does this threshold predict the future risk of type 2 diabetes?
There is no consensus on which HOMA-IR cut-off best distinguishes insulin resistance, so we decided to use the 75th percentile of HOMA-IR assessed a general European population of 2246 nondiabetic adults as the cut-off point (https://doi.org/10.1016/j.diabres.2011.07.015). We have added the reference to the manuscript in line 211. Using a median split (HOMA-IR 2.67) gives the same results (p for difference in postprandial FGF21 response = 0.004).
Was any physical activity data collected in the original study? If so, were the groups compared concerning their PA level at baseline or at follow-up? Clarifying this might be important to ensure that the differences in FGF21 dynamics (or lack thereof) are not affected by physical activity. If no data is collected, please briefly discuss this in the weaknesses.
Physical activity data were not collected in the original study. Indeed, since exercise affects plasma FGF21, it would have been nice to adjust for baseline PA level and/or change in PA. However, because this was a randomised study, we do not expect that baseline PA level differed between the groups. We instructed participants to maintain their habitual PA level throughout the study and participants were unaware of the high-nutrient-quality vs. low-nutrient-quality distinction, so therefore we do not expect differential changes in PA in the intervention groups. We added the lack of physical activity data in lines 376-383:
“Lastly, we did not assess physical activity level and therefore cannot rule out that differences in physical activity level at baseline or follow-up may have affected plasma FGF21 levels, given that exercise modifies circulating FGF21 levels [9]. However, we consider it unlikely that the intervention groups differed in physical activity level at baseline or fol-low-up, because 1) interventions were randomly allocated, 2) participants were unaware of the high-nutrient-quality vs. low-nutrient-quality distinction, and 3) participants were instructed to maintain their habitual physical activity level throughout the study.”
It is interesting to see a negative association between FGF21 levels (fasting and post-prandial) and markers of glucose metabolism in the control group. What does the author think about this finding?
Indeed, we observed a negative correlation between change in both fasting and postprandial FGF21 and change in HbA1c after 12 weeks in the control group, as well as a positive correlation between change in fasting FGF21 and change in glucose and a borderline sign. negative correlation between change in fasting FGF21 and change in liver fat content. These correlations, however, were caused by outliers/data from 2 subjects and excluding these 2 data points resulted in loss of significance and therefore are not discussed further in the manuscript. We have added this to the manuscript in lines 230-232:
“These latter three correlations, however, were driven by data points from two participants, and exclusion of these data points resulted in a loss of significant correlations (ρ = -0.30 to 0.25, p > 0.13).”
In subsection 3.4. Plasma FGF21 and habitual... Please make sure the figure names in the text and figure match.
We have corrected this in lines 239-242 and the figure caption (lines 246-247).
Optional: I think it would be nice to include some information regarding regulating FGF21 synthesis/secretion by nutrients at the cellular level. This may help readers better understand the relationship between FGF21 levels and nutrient intake/sweet-taste preference.
Thank you for this suggestion. We have added some additional information on this in lines 308-313:
“During fasting, liver-derived FGF21 is primarily regulated by the transcription factor PPARα , and intake of fat, protein or a mixed meal results in diminished PPARα-mediated FGF21 secretion, possibly (partly) via a reduction in plasma FFA levels [15,45,46]. High intake of pure simple sugars or alcohol forms an exception; these nutrients acutely elevate plasma FGF21 [30,47-49], likely via activation of hepatic ChREBP [29,50].”

Round 2
Reviewer 2 Report
Thank you for addressing my previous concerns and questions.
Author Response
Your comments and questions were highly appeciated, thank you again.